# Mechanical, Microstructure, and Corrosion Characterization of Dissimilar Austenitic 316L and Duplex 2205 Stainless-Steel ATIG Welded Joints

**DOI:** 10.3390/ma15072470

**Published:** 2022-03-27

**Authors:** Kamel Touileb, Abdeljlil Chihaoui Hedhibi, Rachid Djoudjou, Abousoufiane Ouis, Abdallah Bensalama, Albaijan Ibrahim, Hany S. Abdo, Mohamed M. Z. Ahmed

**Affiliations:** 1Department of Mechanical Engineering, College of Engineering in Al-Kharj, Prince Sattam Bin Abdulaziz University, P.O. Box 655, Al-Kharj 16273, Saudi Arabia; a.hedhibi@psau.edu.sa (A.C.H.); r.djoudjou@psau.edu.sa (R.D.); a.ouis@psau.edu.sa (A.O.); i.albaijan@psau.edu.sa (A.I.); moh.ahmed@psau.edu.sa (M.M.Z.A.); 2Department of Mechanical Engineering, National Engineering School of Tunis (ENIT), El-Manar University, P.O. Box 37, Belvedere Tunis 1002, Tunisia; 3Department of Electrical Engineering, College of Engineering in Al-Kharj, Prince Sattam Bin Abdulaziz University, P.O. Box 655, Al-Kharj 16273, Saudi Arabia; m.benslama@psau.edu.sa; 4Center of Excellence for Research in Engineering Materials (CEREM), King Saud University, P.O. Box 800, Al-Riyadh 11421, Saudi Arabia; habdo@ksu.edu.sa; 5Mechanical Design and Materials Department, Faculty of Energy Engineering, Aswan University, Aswan 81521, Egypt; 6Department of Metallurgical and Materials Engineering, Faculty of Petroleum and Mining Engineering, Suez University, Suez 43512, Egypt

**Keywords:** ATIG, mixing method design, weld aspect, microstructure, mechanical properties, corrosion resistance

## Abstract

The present work analyzed the microstructure, mechanical, and corrosion properties of a dissimilar activated tungsten inert gas (ATIG) welded 2205 duplex stainless-steel (2205 DSS) plate and AISI 316L austenitic stainless steel (316L ASS) and compared them to conventional dissimilar welded tungsten inert gas (TIG). The mixing design method is a tool used to establish the optimal combined flux to achieve a full-penetrated weld bead in one single pass. A microstructure study was carried out by scanning electron microscopy (SEM). The ATIG and TIG fusion zones revealed a matrix ferrite structure with intragranular austenite, Widmanstätten needles, allotriomorphic austenite at the grain boundaries, and plate-like precipitates free of deleterious phases such as sigma and chi phases or second austinite owing to the moderate heat input provided of 0.8 kJ/mm. Ferrite volume proportion measurements were carried out utilizing the areas image processing software. The average ferrite volume proportion attained 54% in the ATIG weld zone; however, it decreased to 47% for the TIG weld zone. The results showed that the optimal flux composed by 91% Mn_2_O_3_ and 9% Cr_2_O_3_ allowed a full penetrated weld to be obtained in one single pass. However, a double side weld is required for conventional TIG processes. The values of the tensile (599 Mpa), hardness (235 HV), and impact test (267 J/cm^2^) measurements of ATIG welds were close to those of conventional TIG welds. The elaborated flux did not degrade the mechanical properties of the joint; on the contrary, it reinforced the strength property. The width of the ATIG heat-affected zone was narrower than that of TIG welding by 2.6 times, ensuring fewer joint distortions. The potentiodynamic polarization test results showed a better electrochemical behavior for ASS 316L than with the weldment and the parent metal of DSS 2205.

## 1. Introduction

Nowadays, the industry has increasingly opted for high-performance materials and processes, which contribute to economic development. Dissimilar welding is a technical feat that meets the expectations of manufacturers. Dissimilar welding is beneficial on two aspects. First, it is beneficial from an economic point of view. Dissimilar welding allows the joining of expensive materials characterized by good corrosion resistance and good mechanical properties with another, cheaper material but of lower quality without altering the soundness of the joint, and it must meet the requirements of strength and safety of the structure being joined [1,2,3]. Secondly, a judicious choice of materials to be welded allows the targeted mechanical properties and corrosion resistance to be reached, needed for specific industry applications such as the nuclear industry, pipelines in desalination plants, and offshore oil and gas pipelines [4,5].

DSS grades are predominately used in marine fabrication industries such as the fabrication of ocean mining machinery, desalination plants, chemical tankers in ships, offshore concrete structures, pipelines, and oil and gas separators [6]. DSSs are usually selected where both high strength and corrosion resistance are required. DSS 2205 has been developed over the last two decades by metallurgists to obtain a stainless steel that is not only stronger than standard marine grade 316 but also offers much better corrosion resistance. The corrosion resistance of DSS 2205 is almost twice that of the ordinary 316. This is due to added levels of nickel, chromium, molybdenum, and nitrogen [7,8]. DSS 2205 is a combination of a ferritic phase with a BCC crystal structure and austenitic phase that crystallizes in FCC with a balance around 50%:50%. The ferrite structure assures high strength and withstands perfectly to stress corrosion cracking (SCC), while the austenite ensures good ductility and general corrosion resistance [9,10].

Austenitic stainless steels remain the most popular grades due to their unique combination of high weldability, high strengthening ability, high ductility, high toughness even at extra-low temperatures, and high corrosion resistance. The good weldability of austenitic stainless steel is affected by the hot cracking that occurs during the welding operation. This defect is ascribed to the presence of a susceptible chemical composition (low-melting-point constituents such as sulfur) and a high level of restraint or tensile stresses present in the weld [11].

The difficulties arising while welding dissimilar materials together are related to the fact that the larger the difference in melting point, the difference in linear expansion coefficient, and the thermal conductivity, the more difficult it is for the weld and the two base metals to meet the requirement of equal strength. To overcome these concerns, metal fillers are chosen related to materials to be joined [12,13], and when required, post-weld heat treatment (PWHT) is used [14] to reduce the mechanical properties to the targeted values. In other cases, the preheat is carried out to avoid any drawbacks in the weld, as in the work conducted by Ghosh et al., to achieve satisfactory joining between AISI 304 austenitic stainless steel to AISI 420 martensitic stainless steel [15].

The joining of austenitic stainless steel and duplex stainless steel is widespread in marine applications due to their combination of better corrosion resistance and good mechanical properties. However, problems such as the hot cracking tendency, the formation of a secondary austenite (γ_2_) phase, particularly in multiple weld passes [16,17], or the appearance of detrimental intermetallic phases such as sigma (σ), chi (χ), and chromium nitride (Cr_2_N) seriously affect the mechanical properties and corrosion resistance of the joint [11,18,19,20,21,22].

Tungsten Inert Gas (TIG) is the most widely used fusion welding process in industry due to its flexibility, on the one hand, and, on the other hand, a variety of materials can be welded using this technique. Unfortunately, this technique allows only 3 mm of depth to be reached in one single pass. In addition, TIG welding is very sensitive to microchemistry variations from cast to cast [23]. Activated Tungsten Inert Gas (ATIG) is an alternative method. The ATIG process was developed at the E.O. Paton Institute of Welding in Kiev in the 1960s. In ATIG welding, a thin layer of flux powder is deposited on edges to be joined. With this technique, 7 mm can be achieved in one single pass without filler metal or edge preparation. ATIG meets the requirements of manufacture in enhancing the productivity [24,25]. The improvement of penetration is related to the following mechanisms. The first mechanism is related to reverse Marangoni convection [26,27]. The second one is related to constriction of arc welding [28,29].

Many studies have investigated the dissimilar DSS 2205 and ASS 316L weld bead by testing different filler metals. For instance, Dhananjay et al. [30] tested different filler metals ER316L and ER309L. They showed in this study that the maximum average tensile strength of 548 MPa was obtained while ER316L filler metal was used compared to the value of 544 MPa when ER309L was tested. In another study, Verma et al. investigated the effects of different filler metals such as E2209, E309LMo, and E309L on dissimilar DSS 2205 and ASS 316L weld beads using multiple weld passes. The obtained tensile strength values were 554 MPa, 544 MPa, and 532 MPa for E2209, E309LMo, and E309L, respectively [31,32].

Limited works have been dedicated to establishing the effects of single flux powders on welds [33], and even fewer have been interested in the effect of combined powders on dissimilar DSS 2205 and ASS 316L weld beads. The novelty of this paper is in the use of the design of experiments mixing method tool to determine the best combination from the available oxide powders as an optimal flux in order to achieve a full penetrated dissimilar DSS 2205 and 316L ASS weld in a single pass without edge preparation, without altering mechanical properties and corrosion resistance of the joints, and without resorting to post-weld heat treatment (PWHT).

## 2. Materials and Methods

### 2.1. Material

The materials used in this study were the austenitic stainless-steel grade 316L and the duplex stainless-steel grade 2205 in rectangular plates of 6 mm thickness and 200 mm length × 100 mm width. The chemical composition of both stainless steels is given in Table 1. Table 2 depicts the oxides used and their physical properties.

### 2.2. Welding Procedure

ATIG welding was used to weld austenitic SS 316L and duplex SS 2205 in dissimilar butt joints. Before welding, the plates were cleaned with acetone. To eliminate humidity of the powder, it was heated separately in a furnace at 100 °C for 1 h. A thin layer of a mixed powder with methanol in the proportion of (1:1) made in a form of paste was applied using a brush on plain edges to be joined. The mean coating density of flux was about 4–5 mg/cm^2^. The joints were executed with a square butt weld design without edge preparation. Both plates were clamped with zero gap distance, as shown in Figure 1. The welding parameters used are reported in Table 3. The heat input recommendation is 0.5 to 2.5 kJ/mm, so the welding parameters chosen provided an acceptable range of moderate heat input of 0.8 kJ/mm.

After the ATIG welding, the samples were cut from the welded joints far from the welding starting point to be sure that the arc welding was stabilized, as shown in Figure 2.

### 2.3. Design of Experiments Methodology

Design of Experiments (DOE) is considered one of the most essential statistical tools for designing high-quality experimental systems at a reduced cost. In this study, the mixing method was used and Minitab 17 software was the most appropriate tool for this purpose. In the first step, eight kinds of oxides (SiO_2_, TiO_2_, Fe_2_O_3_, Cr_2_O_3_, ZrO_2_, Mn_2_O_3_, V_2_O_5_, CoO_3_, MgO) were tested. Single oxides were deposited on both materials and welding operation was carried out. Among these eight oxides, three oxides candidates Cr_2_O_3_, Fe_2_O_3_, and Mn_2_O_3_ that gave the best depth of penetration and high ratio were selected to be used in the mixing design method. In the second step, based on the simplex lattice degree of four designs, nineteen combinations from the selected oxides were prepared. Finally, the optimal combination obtained was 91% Mn_2_O_3_ and 9% Cr_2_O_3_. Finally, a conventional TIG welding line and another with the ATIG technique were carried out. 

### 2.4. Microstructure Investigation

The microstructural evolution of the fusion zone of both TIG and ATIG welds was investigated using a JEOL JSM-7600F scanning electronic microscope (SEM). Before investigation, the samples were polished up to 1200 grit fineness, followed by a cloth polishing to a 0.05 μm alumina surface finish. Then, samples were etched using Glyceregia solution (15cc HCl + 5cc HNO3 + 10cc glycerol). The areas image processing software from Microvision Instruments was used to measure the ferrite volume proportions.

### 2.5. Tensile Test

The tensile tests were carried out using a computer control electrohydraulic servo universal testing machine model WAW-300E at a cross head speed of 0.5 mm/min, a loading rate of 0.5 kN/s, and a strain rate of 1.6 × 10^−4^ s^−1^. The tensile tests were conducted for both dissimilar TIG, dissimilar ATIG, 316L/316L butt joints, and 2205 DSS/2205 DSS butt joints. For each sample category, three samples were prepared and tested. The tests were conducted according to the requirements of ASTM E8M-04, as shown in Figure 3.

### 2.6. Hardness Test

Micro Vickers hardness tests were performed by a digital hardness tester model HVS-50 with a standard load of 100 gf and dwell time of 10 s. The test was conducted according to ASTM E-384-99. The hardness line and tracks indentation are displayed in Figure 4. The measurements were performed on each sample with about 0.5 mm between two indentations. The hardness line measurements were far from the top surface by 2 mm.

### 2.7. Impact Test

For impact tests, 3 samples for the dissimilar TIG welds and 3 samples for the dissimilar ATIG welds were prepared according to the ASTM E23 standard with the dimensions shown in Figure 5. A Charpy “V” notch impact testing machine was used for impact testing.

### 2.8. Corrosion Behavior

Corrosion tests were carried out using a potentiodynamic polarization electrochemical system to obtain the main corrosion parameters such as corrosion potential (Ecorr), pitting potential (Ep), and corrosion current density (icorr) that can be used to evaluate the corrosion resistance. Electrochemical tests were performed using a potentiostat system of AUTOLAB-PGSTAT302N. Before testing, the samples were cut to the dimensions of 20 mm × 10 mm and grinded up to 1200 grit with SiC emery papers, taking into account six different regions of the welded joint: TIG weld metal, ATIG weld metal, TIG ASS 316L HAZ, ATIG ASS 316L HAZ, TIG DSS HAZ, and ATIG DSS HAZ, in addition to ASS 316L and DSS base metals. The data were collected after immersion in 3.5% NaCl solution for 1 h at room temperature. The tests were performed in the 3.5% NaCl solution at a scan rate of 1 mV/s. Platinum (Pt) was used as the auxiliary electrode, silver chloride (Ag/AgCl) was used as the reference electrode, and the sample was used as the working electrode.

## 3. Results and Discussions

### 3.1. Weld Bead Aspect

#### 3.1.1. Selection of Candidate Oxides

The joints were executed with the butt weld design without edge preparation. Both plates were clamped with zero gap distance. Single oxide flux was deposited on plates to be welded with a total width of 10 mm. TIG weld and nine ATIG welds were carried out. The results in Table 4 clearly display that the three oxides candidates for the next steps are Cr_2_O_3_, Fe_2_O_3_, and Mn_2_O_3_ because they exhibit the highest penetrated weld beads and ratios (7.47 mm-1.66, 8.66 mm-1.68, and 7.02 mm-1.63), respectively.

#### 3.1.2. Mixture Contour of Plot

According to the mixture method, simplex lattice degree four is the most appropriate for our experiments. We prepared nineteen compositions with different proportions of oxides selected, which were Fe_2_O_3_, Cr_2_O_3_, and Mn_2_O_3_. Table 5 shows the chemical compositions of the nineteen combinations and the related results of depths and ratios.

The compositions of the flux were the input data, and both the depth (D) and the ratio (R) were the output response. To visualize the relationships between the components in a three-component mixture, the triangular coordinate systems were used. Figure 6 shows the contour plots for depth, and the ratio that generated based on the experimental results is shown in Figure 7.

From the mixture contour plot for depth D, three main regions can be observed, where the maximum depth can be attained. The first region is close to manganese oxide, the second region is close to chromium oxide, and the third one is far from the base of the triangle, as can be seen in Figure 6.

The mixture contour plot for flux ratio shows three main regions where the maximum depth can be achieved. The first region is close to manganese oxide, the second region is close to chromium oxide, and the third one is close to the triangle rib (Cr_2_O_3_–Fe_2_O_3_) from the base of the triangle, as shown in Figure 7.

The optimizer module available in Minitab 17 software waws used to find the optimal composition. Figure 8 shows the optimization plot that indicates how the variables of Fe_2_O_3_, Cr_2_O_3_, and Mn_2_O_3_ affect the predicted responses in terms of both penetration depth D and flux ratio R. It should be noted that the numbers at the top of the columns show the current variable settings and the high and low variables settings in the data. Two points for each cell represent the two levels of the categorial variable. If the level for each variable is equal to 1, it indicates a high level, and if it is equal to 0, it indicates a low level. The level between the high and low levels represents the best mixing flux composition, which is 91% Mn_2_O_3_ + 9% Cr_2_O_3_ + 0% Fe_2_O_3_.

Figure 8 shows the predicted response “y” and the individual desirability score “d” for both the depth and ratio for the current variable settings, as can be seen from the first column. Hence, the predicted response for the depth is y = 8.75 mm and the corresponding desirability is 0.99. The predicted response for the ratio is y = 1.76 and the corresponding desirability is 0.97. Thus, the overall composite desirability becomes 0.98. This indicates that the variables achieve favorable results for all responses, and it means that both responses are within acceptable limits.

#### 3.1.3. Validation Test

The third step is about the validation weld line. Accordingly, the combination flux was prepared. ATIG dissimilar welding was executed using the optimal composition with the same conventional parameters of the TIG weld. The transverse cross-sections of the weld beads were investigated using an optical microscope CAROLINA (CAROLINA, Burlington, NC, USA). Based on that investigation, the obtained penetration depth (D) is 8.93 mm and the bead face width (W) is 8.87 mm for ATIG welding, which lead to an aspect ratio D/W of 1.94. It can be noted that the depth is increased by about 2.4 times and the ratio is enhanced by about 5.7 times compared to the conventional TIG welding. Better yet, the measured depth and ratio are higher than the predicted values of 8.75 mm and 1.75, respectively. The depth bead profile data of the conventional TIG weldments and of ATIG with the optimal flux are listed in Table 6. Figure 9 presents macrographs for the cross-section of the TIG welded bead and those of the ATIG with the optimal flux. It is clearly observed that the bead has a full penetration after ATIG welding.

### 3.2. Microstructure Assessment

Based on a WRC-1992 diagram before welding, the DSS Creq/Nieq ratio equals 2.63. DSS solidifies in ferrite mode. The DSS microstructure consists of both ferrite (α) and austenite (γ) phases in almost equal proportion. However, the ASS 316L Creq/Nieq ratio equaling 1.56 solidifying in austenite-ferrite mode leads to the formation of austenite (γ) with the formation of strip δ ferrite. 

The weld bead microstructure shown in Figure 10a for the ATIG weld and Figure 10b for the TIG weld indicates the existence of delta ferrite (δ) and austenite (γ) phases. First, the solidification of molten metal results in the formation of ferrite matrix immediately after solidification, and then the nucleation of austenite phases starts upon a further cooling cycle. In the weld region, three different austenite phases’ morphologies are found. They are a thin grain boundary austenite, an elongated Widmanstätten austenite structure, and an intragranular austenite phase. Transverse profile ferrite volume proportions of the 2205/316L stainless-steel dissimilar welded joint are depicted in Table 7 and shown in Figure 11. They reveal an evolution of ferrite volume proportions seen from the HAZ of the 316L ASS to the 2205 DSS HAZ. The average ferrite volume proportion in the ATIG weld zone reaches 57%, greater than that of the TIG weld zone of 47%.

The ATIG weld with optimal flux increases the arc voltage, and the amount of heat input per unit length in a weld is also increased. The energy density of the source leads to high heat input. However, the arc weld of TIG welding without flux has a lower energy density and, therefore, in this case, a lower heat input in comparison with ATIG. TIG welding associated with low heat input is characterized by a relatively rapid cooling rate in comparison to that of the ATIG weld [34]. In the ATIG weld zone, a slow cooling rate allows for more transformation of ferrite to different types of austenite (GBA (ferrite grain boundaries), Widmanstätten, IGA (Intragranular grain austenite)). However, the TIG weld specimen executed using double passes receives more heat transferred to the joint and, consequently, more formation of austenite is expected, as well as higher ferrite volume proportions that influence the mechanical properties and reinforce the strength and hardness properties [35].

For the TIG weld, the HAZ width of about 300–500 μm is observed on the duplex steel side, as shown in Figure 12a. The HAZ microstructure consists of lamellar austenite precipitates located mainly on the equiaxial high ferrite grain boundaries and, in a minor amount, inside the ferrite grains with the presence of small proportions of Widmanstätten side plates off the ferrite grain boundary. The ferrite volume proportion is up 69% in that zone, which is significantly higher in comparison to the parent material of 2205 DSS. The wide HAZ of about 120–160 μm is observed on the 316L ASS side, as shown in Figure 12b. The microstructure consists of strips of ferrite precipitates that surround austenite grains. In the 316L ASS HAZ side, the ferrite volume proportion is up to 8%, as listed in Table 7. The width of the DSS HAZ side is higher than that of 316L HAZ by 2.5–3.1 times. This result is in good concordance with that obtained in the study of Topolska et al. [36].

In the ATIG weld, the HAZ of about 120–200 μm is observed on the duplex steel side, as shown in Figure 13a. Moreover, the microstructure consists of continuous networks of austenite at the ferrite grain boundaries (GBA) and also the austenite phase formed as Widmanstätten (WA), in addition to intragranular austenite within the ferrite grains (IGA). The ferrite volume proportions in this zone reaches 63%, which is significantly higher compared to the parent material of 2205 DSS.

The wide zone of about 60–80 μm is observed on the HAZ 316L side, as shown in Figure 13b. The ATIG HAZ DSS side is wider than that of the TIG HAZ 316L side by 2 to 2.5 times. The microstructure consists of δ ferrite precipitates in a matrix of austenite grains. The ferrite volume proportions in this zone decrease to 6.9%. We notice that the TIG HAZ is wider than that of ATIG HAZ for both sides, respectively. This effect can be ascribed to the fact that the heat provided to the ATIG weld bead is more concentrated owing to constriction of the ATIG arc weld than that of the TIG weld [37].

On the other hand, the HAZ of the DSS side for both ATIG and TIG welds is wider than that of the 316L side, owing to a higher thermal conductivity of 2205 DSS compared to that of 316L. This result is in good agreement with the results obtained by Taheri et al. [35].

### 3.3. SEM/EDS Analysis

Referring to Figure 14a, we notice an increase in the Cr element at the fusion border of the 316L side in the TIG weld. This observation clearly indicates the presence of more δ ferrite phase in this region compared to the ATIG weld shown in Figure 14b.

We also remark, in the latter region, the presence of the Mo element close to that of 316L parent metal that suggests that the ATIG weld could withstand aggressive environments.

Cr, Ni, Mn, and Mo element contents labeled in Figure 15b of the ATIG DSS 2205 side fusion border are close to the parent metal, leading to more ferrite phase formation. However, in Figure 15a, we remark an increase in the austenite phase, promoting elements such as Ni and Mn, which indicates the formation of more austenite phase compared to ATIG DSS 2205.

### 3.4. Tensile Test

The average value of the ultimate tensile strength (UTS) of the ATIG dissimilar weld is 599 MPa, which is close to that of conventional TIG welding (594 MPa), as shown in Table 8.

We notice that the fractures for ATIG and TIG dissimilar welds are localized at the base metal, precisely at the ASS 316L base metal and outside the weld zone, as shown in Figure 16. This may be ascribed to differences in the chemical composition that promote the dual-phase nature and a stronger solid solution strengthening mechanism as mentioned in some works [38,39]; hence, a reduction in ductility of these zones must be expected. The DSS BM has a strength superior by 205 MPa of that of 316L BM. Indeed, the duplex material contains the ferrite phase, which allows a material with greater UTS. 

The fracture faces in both cases show similar ductile elongated large dimples, attesting the reduction in ductility of 316L austenitic stainless steel with cleavage facets, as shown in Figure 17.

The obtained strength does not denote the strength of the weld joint. The weld joint strength is higher than the value obtained in TIG and ATIG welds. The TIG specimen fractured at 594 MPa is about 96% of the 316L base metal strength. In addition, the ATIG specimen breaks at 599 MPA, which is about 97% of the 316L base metal strength. The test shall be accepted because it meets the requirements. The values of the standard deviations (σ) are less than 5 MPa, which attests the accuracy and reliability of the obtained results. Figure 18 displays the tensile test curves of the welds, which are similar TIG 316L (#1), similar DSS 2205 (#2), dissimilar ATIG (#3), and dissimilar TIG (#4). The used flux does not negatively affect the tensile property of the weld.

### 3.5. Hardness Test

Variations in the Vickers micro hardness as a function of the distance from the DSS 2205 to the 316L SS in the sample are shown in Figure 19 for both ATIG and TIG welds. The hardness at the weld zone of the ATIG weld (average 235 HV) shows little variation compared to that of the TIG weld (237 HV), which is more constant.

The average hardness value in the ATIG fusion region (235 HV) is roughly equal to that of the TIG fusion zone (237 HV) weld. The highest value of hardness (242 HV) is located at the TIG weld zone at the nearest point of DSS/WZ. The standard deviations are less than 4 HV, which indicates good hardness homogeneities in the joints, as shown in Table 9.

The results collected in Table 10 are related to the micro hardness tests at the HAZ of the weldments. The average hardness value is found to be greater at the HAZ of the DSS 2205 side (257.5–257 HV) compared to that of the 316L side (222–210 HV) for dissimilar TIG and dissimilar ATIG welds. On the other hand, the standard deviations are less than 4 HV, which attests the small disparities in the obtained hardness values between the maximum and the minimum values.

### 3.6. Impact Test

The impact tests were carried out only in the fusion zones of ATIG and TIG welds. The obtained experimental results are displayed in Figure 20. The average energy absorbed in the fusion zone in the case of the ATIG 2205 DSS/316L SS dissimilar weld (267 J/cm^2^) is slightly lower than that of the dissimilar TIG weld (275 J/cm^2^). The dissimilar ATIG and dissimilar TIG welds have almost the same ability to withstand shocks. The standard deviations are less than 8 J/cm^2^. Figure 21 shows the broken TIG and ATIG dissimilar weld specimens after the impact test.

The SEM fractography represented in Figure 22a belongs to the TIG weld. It indicates multiple and fine dimples with ductile dimple tearing resulting from the coalescence of micro voids, which confirms the ductile fracture mode during the impact test. Figure 22b shows the formation of a network of multiple homogenous dimples without voids, which characterizes the ductile fracture for the ATIG weld. The absorbed energy of the ATIG weld is close to that of the TIG weld, which ensures that the elaborated flux does not adversely affect the resistance of the ATIG weld during sudden loads.

### 3.7. Corrosion Behavior

The potentiodynamic polarization curves of the base metals, heat-affected zones, and weld metals in the 3.5% NaCl solution at room temperature are illustrated in Figure 23 and Figure 24. The corrosion current (Icorr), the corrosion potentials (Ecorr), and the pitting potentials (Epit) of the different samples are summarized in Table 11. It can be seen that the weld metals display a passive behavior similar to those of the ASS 316L and DSS 2205 base metals. However, the anodic behavior of the weld metal in TIG and ATIG are notably different from the ASS 316L and DSS base metals, both in terms of the corrosion potential (Ecorr) and current density, which can be attributed to the difference in the chemical composition and microstructure. The parent metal of ASS 316L exhibits a minimum corrosion rate when compared to DSS 2205, TIG, and ATIG weldment. ASS-BM exhibits a corrosion potential of −927 mV versus SCE, which is more notable than the potentials observed for DSS-BM, as well as weld metals TIG and ATIG (−974 mV, −1050, and −1032 mV), respectively. On the other hand, Table 11 shows the values of icorr for both the TIG weld zone (54.5 μA/cm^2^) and ATIG weld zone (54.1 μA/cm^2^), and those for TIG and ATIG HAZ in both ASS and DSS regions are higher than those of the base metals (24.4 μA/cm^2^ and 29.8 μA/cm^2^ for ASS and DSS, respectively). This indicates that protection of the passive film formed in these regions is less than that of the base metals. This can be explained by the deformation resulting from the microstructural changes and the residual stresses generated by the welding thermal cycle [40].

Furthermore, it is also noted that TIG and ATIG weld metals, as well as TIG and ATIG HAZ in DSS region, have values of pitting potentials higher than those of the base metals ASS (−455 mVSCE) and DSS (−434 mVSCE), which means that these regions have better performance with respect to localized corrosion.

On the other hand, it is again observed that TIG and ATIG HAZ in ASS regions display low values of Epit in the solution of 3.5% NaCl, indicating that these regions of the welded joint are more susceptible to localized pitting corrosion.

## 4. Conclusions

In this work, a comparison between dissimilar ATIG and dissimilar TIG welds of DSS grade 2205 steel and ASS grade 316L was carried out. Having investigated the mechanical properties, microstructure, and corrosion behavior of the welds, the following main conclusions can be drawn:−The mixing method design is a key tool to elaborate the flux composed by 91% Mn_2_O_3_ and 9% Cr_2_O_3_. The ATIG weld carried out with the optimal flux achieves a full penetration weld without edge preparation or the use of a filler metal. The depth in the ATIG weld is greater than that of conventional TIG by 2.4 times.−The volume proportion of ferrite, based on areas image analysis, shows that the average ferrite volume proportion in the ATIG weld zone reaches 57%, greater than that of the TIG weld zone (47%). Moreover, the welds are free of deleterious phases such as sigma phases owing to the moderate heat provided to the weld pool of 0.8 kJ/mm.−In the ATIG weld, the 2205 DSS HAZ width is 200 μm, while in the 316L ASS one, HAZ is about 80 μm. In the same way, in the TIG weld, the 2205 DSS HAZ is very wide, attaining 500 μm, while in the 316L ASS one, the HAZ width is 160 μm. The HAZ of 2205 DSS sides contains more ferrite phase, while the 316L sides are mainly composed by the austenite phase with sparse bands of δ ferrite.−The tensile strength value of the ATIG weld metal reaches 599 MPa, which is closer to that of the TIG weld (594 MPa). The hardness measurements at the weld zones shows close results for both TIG and ATIG, with average values of 237 HV and 235, respectively. The average energy absorbed during the toughness impact test reaches 267 J/cm^2^ for the ATIG weld compared to 275 J/cm^2^ for the TIG weld. The elaborated optimal flux has no negative effect on the mechanical properties of the welds.−The weld metals display passive behavior similar to those of ASS 316L and DSS base metal. −The parent metal of ASS 316L was subjected to a minimum corrosion rate when compared to DSS 2205, TIG, and ATIG weldment.−Moreover, TIG and ATIG HAZ in ASS regions display low values of Epit, indicating that these regions of the welded joint are very susceptible to localized pitting corrosion.

## Figures and Tables

**Figure 1 materials-15-02470-f001:**
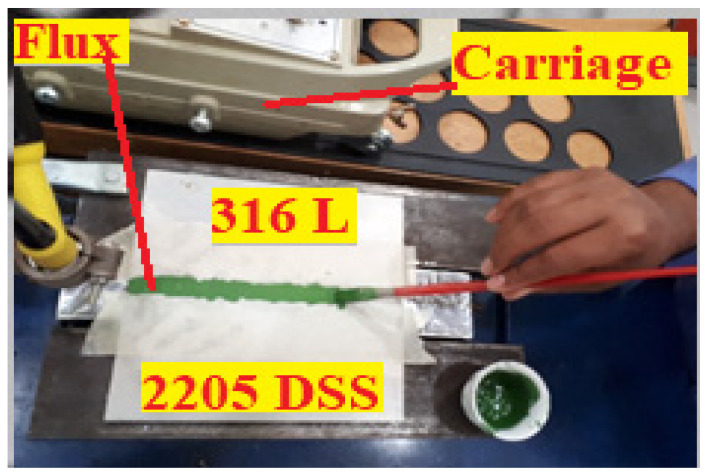
The deposition of the mixed flux on the workpiece before ATIG welding.

**Figure 2 materials-15-02470-f002:**
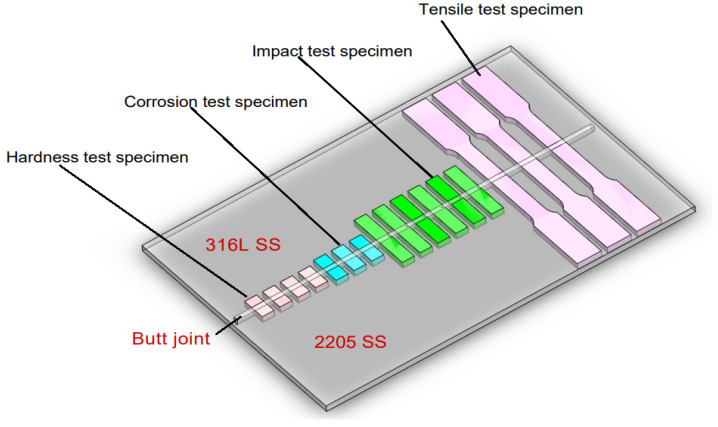
Schematic drawing showing test specimens taken from the dissimilar ATIG welded DSS 2205 and 316L stainless steels for the different types of tests.

**Figure 3 materials-15-02470-f003:**
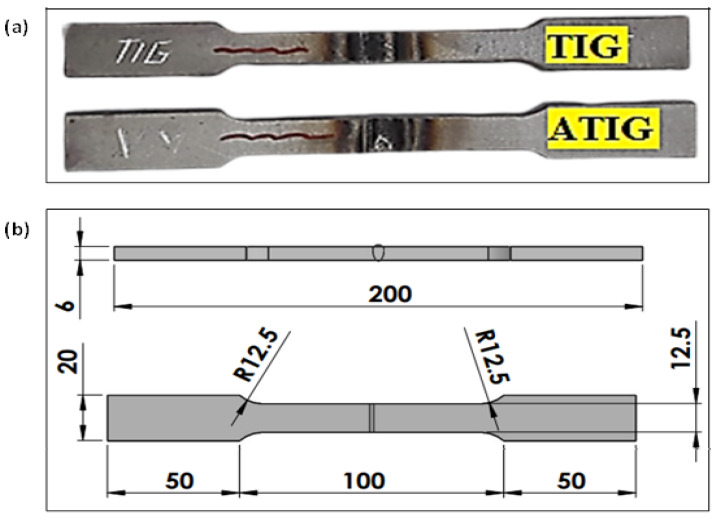
Tensile test specimen (**a**); dimensions of specimen according to ASTM E8M-04 (**b**) (units in mm).

**Figure 4 materials-15-02470-f004:**
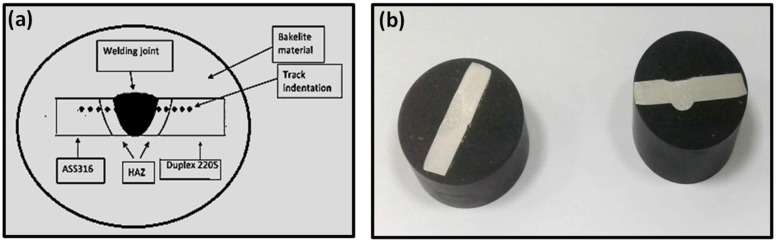
Microhardness test specimens’ indentation locations (**a**) for ATIG samples (**b**) after hot mounting.

**Figure 5 materials-15-02470-f005:**
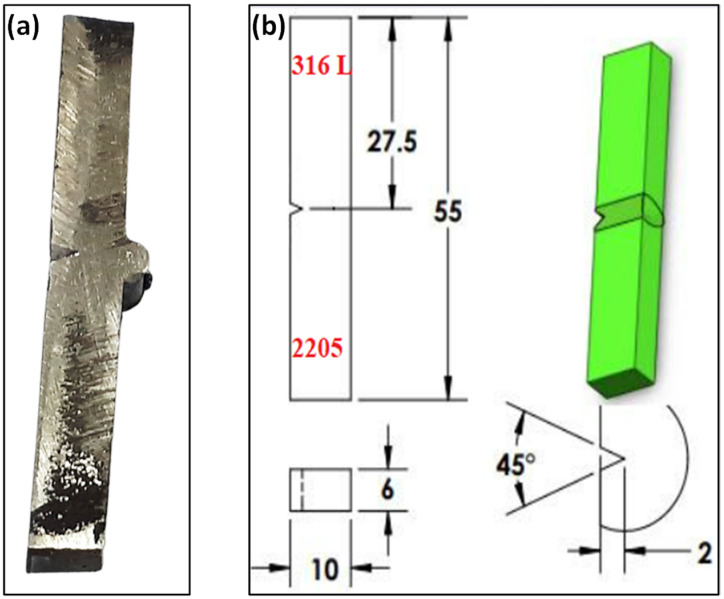
Impact test specimen (**a**); dimensions of specimen (**b**) (units in mm).

**Figure 6 materials-15-02470-f006:**
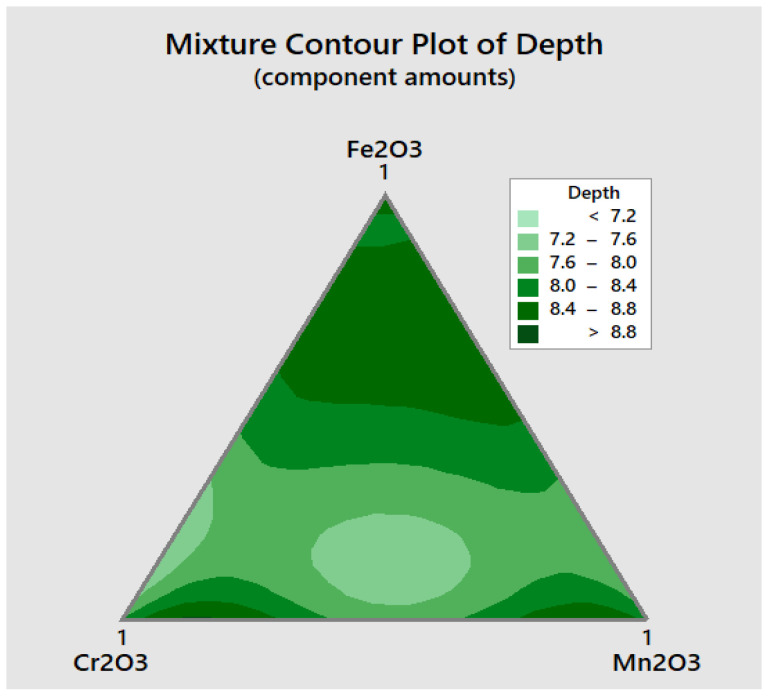
Mixture contour plot for the depth D of penetration.

**Figure 7 materials-15-02470-f007:**
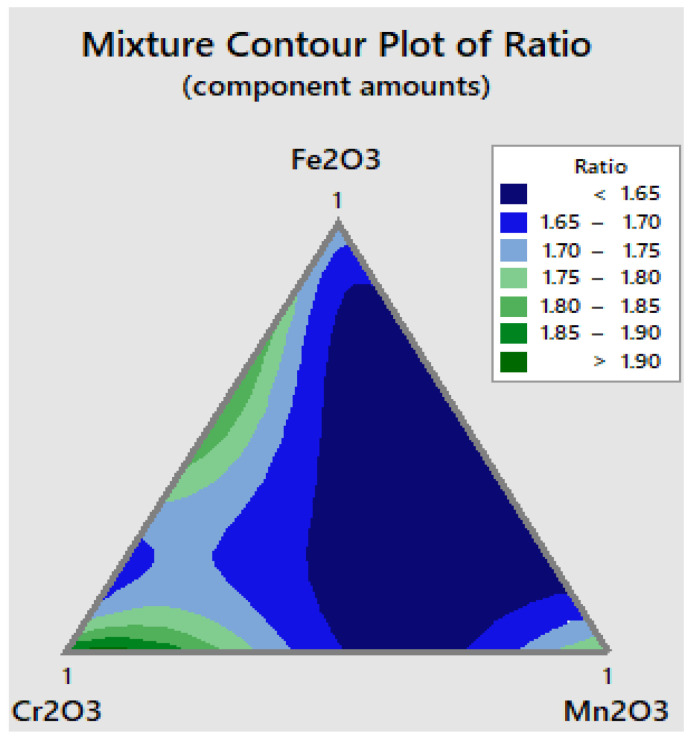
Mixture contour plot for ratio R.

**Figure 8 materials-15-02470-f008:**
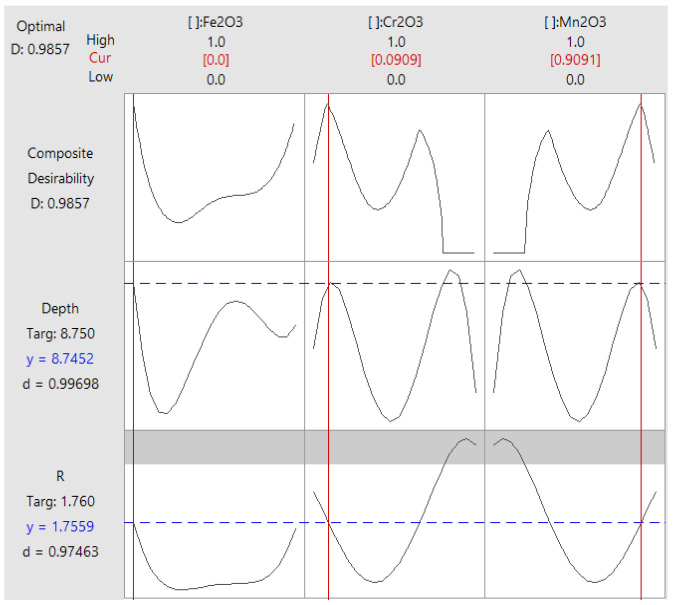
Optimization plot for depth D and ratio R.

**Figure 9 materials-15-02470-f009:**
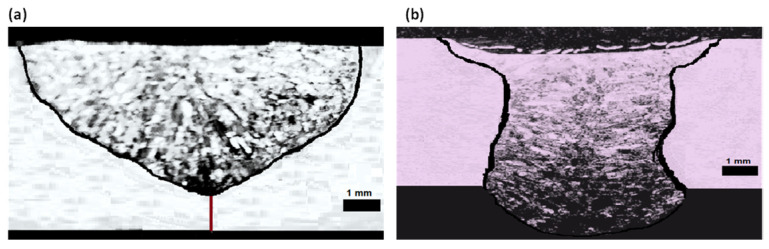
Transverse optical macrographs of the dissimilar welded beads using TIG (**a**) and ATIG (**b**).

**Figure 10 materials-15-02470-f010:**
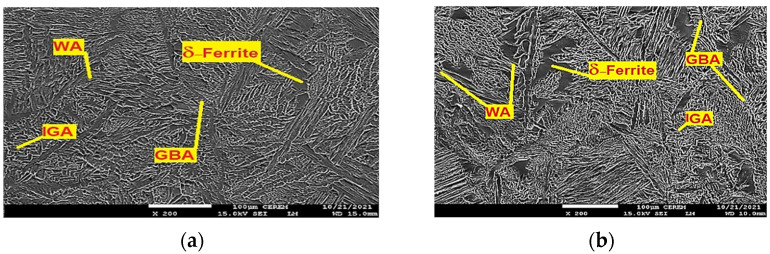
TIG weld zone (**a**) and ATIG weld zone (**b**) (500×).

**Figure 11 materials-15-02470-f011:**
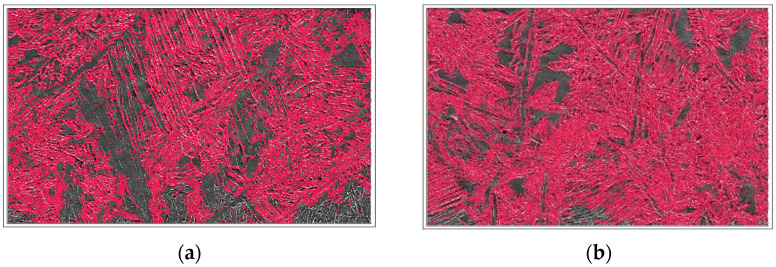
Ferrite proportion measurements of dissimilar TIG (**a**) and ATIG (**b**) welds (500×).

**Figure 12 materials-15-02470-f012:**
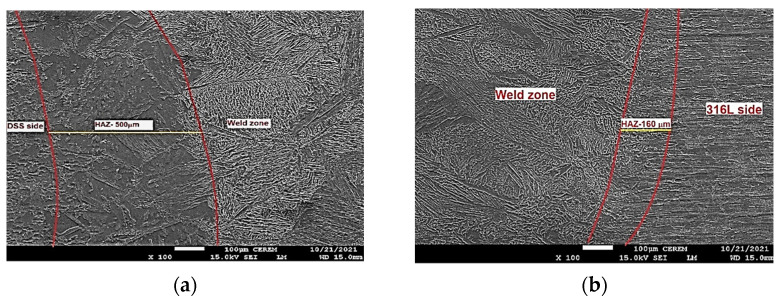
TIG HAZ DSS side (**a**) and 316L side (**b**) (×100).

**Figure 13 materials-15-02470-f013:**
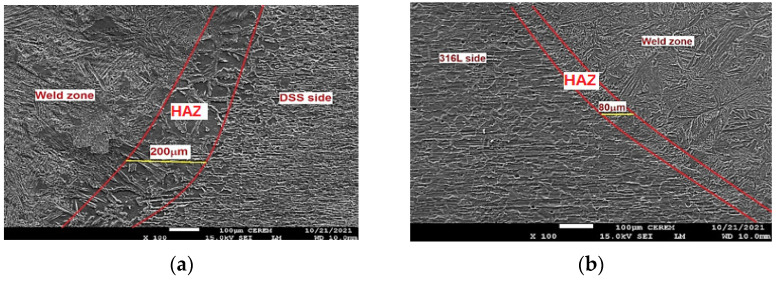
ATIG HAZ DSS side (**a**) and 316L side (**b**) (×100).

**Figure 14 materials-15-02470-f014:**
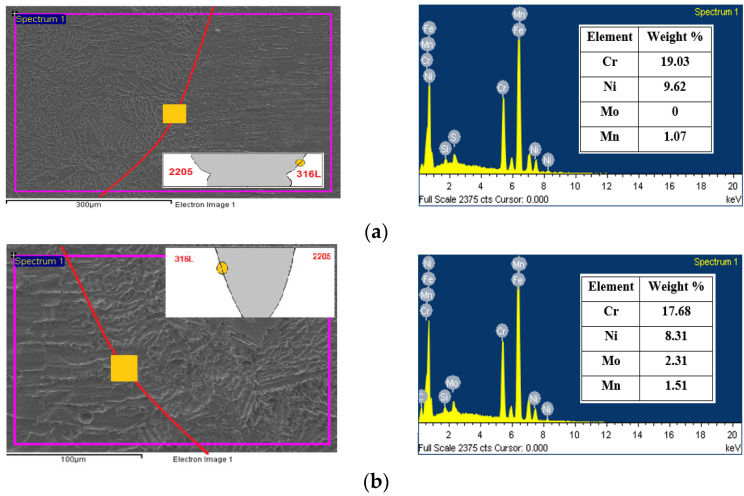
SEM-EDS micrographs and data obtained for TIG (**a**) and ATIG (**b**) ASS 316L side fusion border.

**Figure 15 materials-15-02470-f015:**
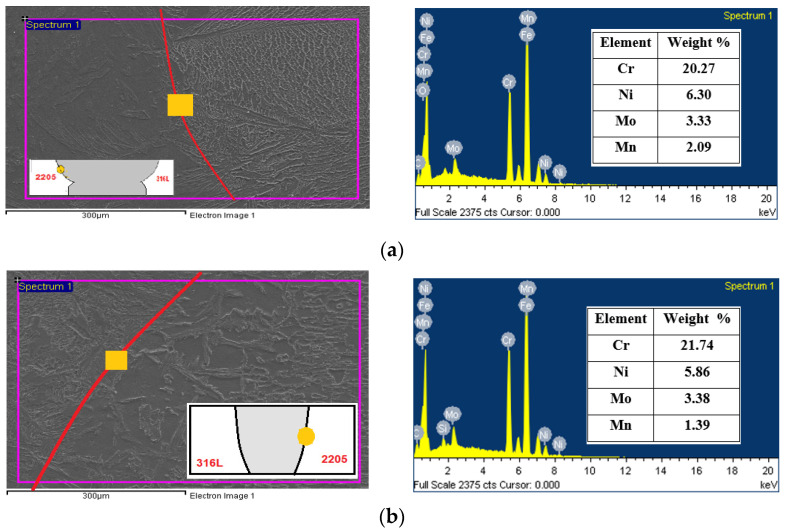
SEM-EDS micrographs and data obtained for TIG (**a**) and ATIG (**b**) DSS 2205 side fusion border.

**Figure 16 materials-15-02470-f016:**
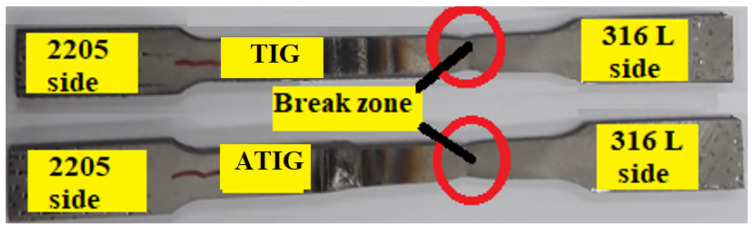
Specimens of the tensile testing after the test.

**Figure 17 materials-15-02470-f017:**
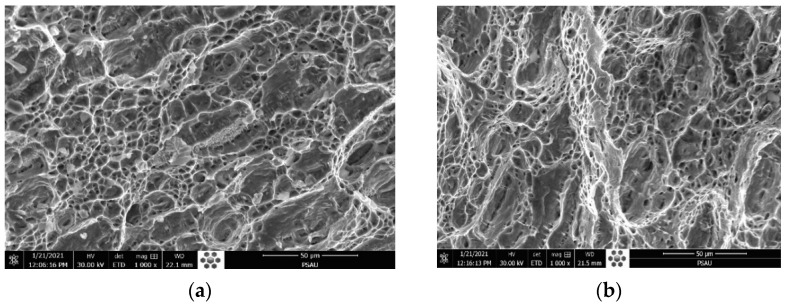
Fracture surface of TIG (**a**) and ATIG (**b**) dissimilar 316L- 2205 DSS at break zone (×1000).

**Figure 18 materials-15-02470-f018:**
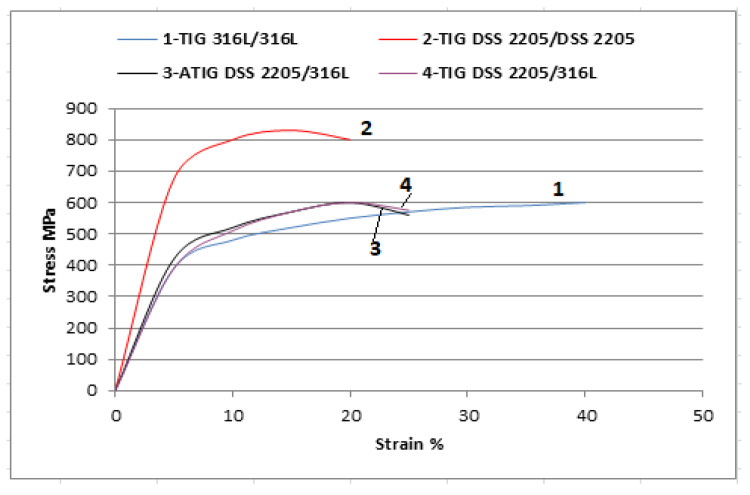
Stress vs. strain curves of the different weldments.

**Figure 19 materials-15-02470-f019:**
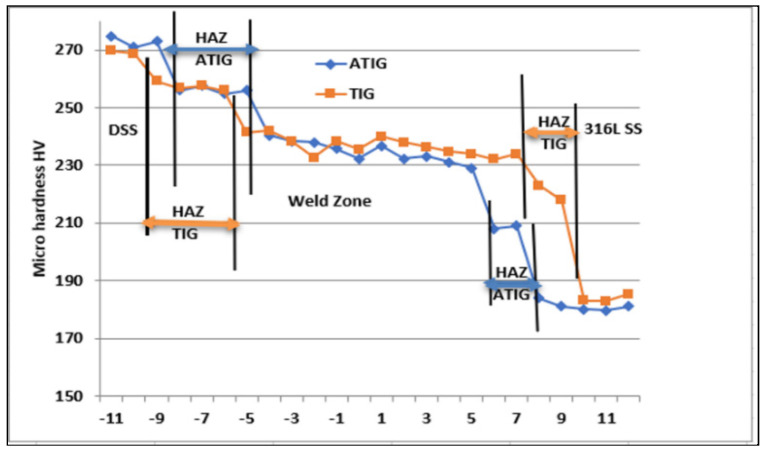
Micro hardness profiles across the centerline of dissimilar ATIG and dissimilar TIG welds.

**Figure 20 materials-15-02470-f020:**
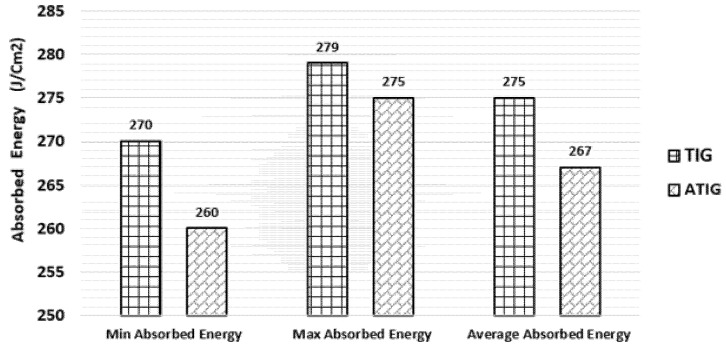
Measurements of energy absorbed of TIG and ATIG (optimal flux) at fusion zone for dissimilar 2205 DSS/316L SS weld.

**Figure 21 materials-15-02470-f021:**
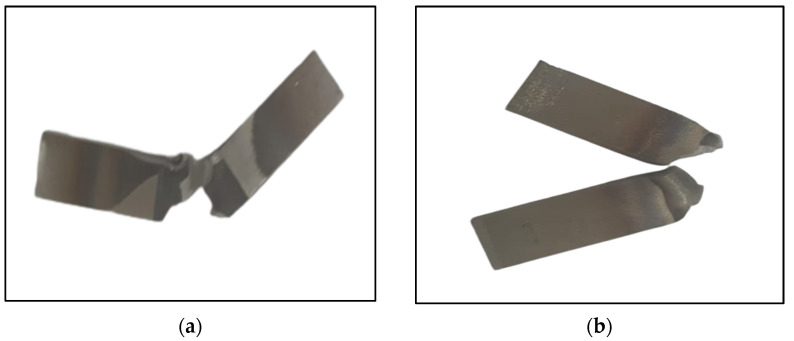
Specimens of TIG (**a**) and ATIG (**b**) after the impact testing.

**Figure 22 materials-15-02470-f022:**
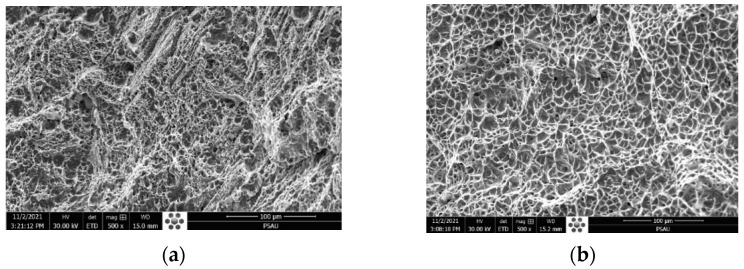
SEM fractography of dissimilar TIG (**a**) and ATIG (**b**) weldments in the FZ (×500).

**Figure 23 materials-15-02470-f023:**
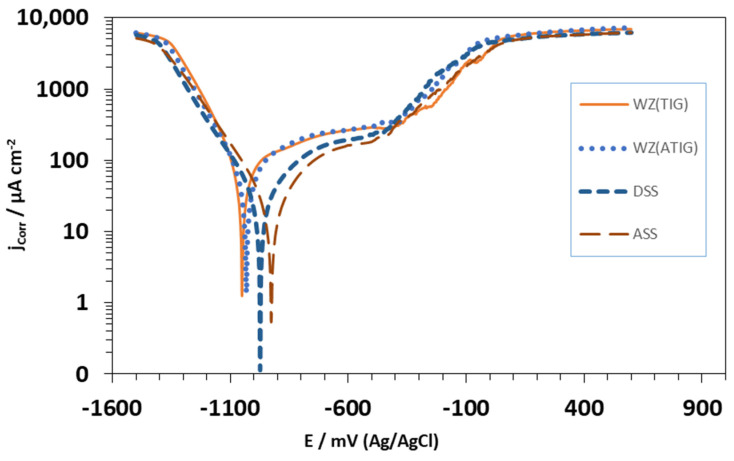
Curve of potentiodynamic polarization for electrolyte 3.5% NaCl solution obtained under the areas of TIG and ATIG weld metal and base metals.

**Figure 24 materials-15-02470-f024:**
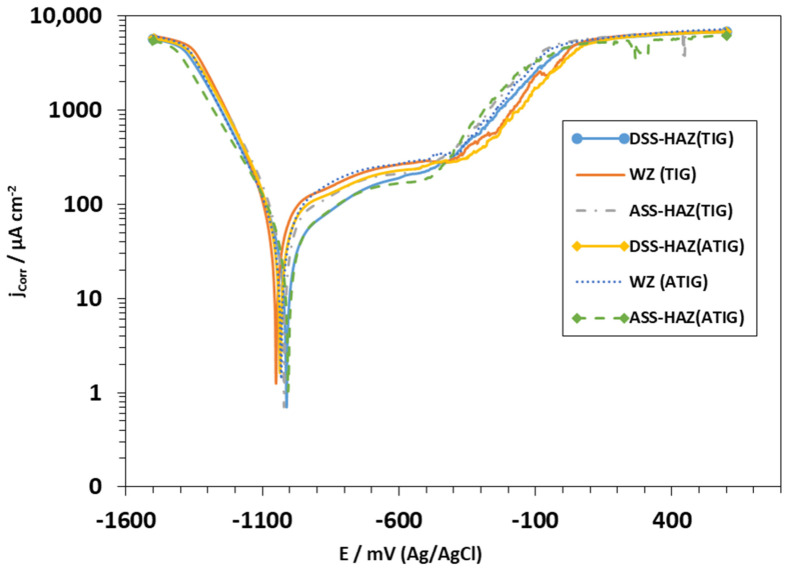
Curve of potentiodynamic polarization for electrolyte 3.5% NaCl solution obtained under the areas of HAZ of the AISI 316L alloy and HAZ of the AISI 2205 alloy for both TIG and ATIG, and again for the weld metal (kept as reference).

**Table 1 materials-15-02470-t001:** Chemical composition (wt.%) of 316L and duplex grade 2205 stainless steels.

Elements	C	Mn	Si	P	S	Cr	Ni	Mo	N	Cu	Co	Fe
316L SS	0.026	1.47	0.42	0.034	0.0016	16.60	10.08	2.14	0.044	0.50	-	Balance
2205 SS	0.016	1.35	0.47	0.025	0.001	22.42	5.71	3.15	0.17	0.21	0.14	Balance

**Table 2 materials-15-02470-t002:** Oxides used and their physical properties.

Oxides	SiO_2_	TiO_2_	Fe_2_O_3_	Cr_2_O_3_	ZrO_2_	Mn_2_O_3_	V_2_O_5_	Co_2_O_3_	MgO
Melting Temperature (°C)	1722	1830	1540	2435	2715	940	690	895	2600
Dissociation energy ΔH° (kJ/mol)	902	941	826	1128	1080	971	1551	577	572

**Table 3 materials-15-02470-t003:** Welding parameters used in this study.

Parameters	Range
Welding voltage	10 Volts
Arc efficiency	75%
Welding current	180 A
Welding speed	110 mm/min
Arc Length	2 mm
Electrode tip angle	45°
Shielding gas on the workpiece	Argon with a flow rate of 12 L/min
Shielding gas on the backside	Argon with a flow rate of 5 L/min
Welding mode	Negative direct current electrode

**Table 4 materials-15-02470-t004:** Weld aspect of single oxides flux of dissimilar ATIG welds.

Oxides	SiO_2_	TiO_2_	Fe_2_O_3_	Cr_2_O_3_	ZrO_2_	Mn_2_O_3_	V_2_O_5_	Co_2_O_3_	MgO
**Depth (mm)**	6.59	5.36	7.47	8.66	4.33	7.02	7.02	7.01	4.16
**Width(mm)**	7.61	10.72	8.43	9.68	8.66	8.01	9.21	9.04	8.32
**Ratio**	1.60	0.50	1.66	1.68	0.5	1.63	1.61	1.57	0.50

**Table 5 materials-15-02470-t005:** Dimensions of depth (D) and R (ratio) of ATIG weld beads.

Exp.	Fe_2_O_3_%	Cr_2_O_3_%	Mn_2_O_3_%	Depth (mm)	R = (D + Wb)/Wf
1	75	25	0	8.78	1.83
2	75	0	25	8.84	1.65
3	50	25	25	8.68	1.64
4	50	0	50	8.27	1.64
5	50	50	0	7.95	1.83
6	25	75	0	7.48	1.7
7	25	50	25	7.3	1.66
8	25	25	50	7.15	1.54
9	0	75	25	8.74	1.84
10	0	50	50	7.75	1.66
11	0	25	75	8.5	1.65
12	25	0	75	7.91	1.61
13	33.33	33.33	33.33	8.79	1.7
14	66.667	16.667	16.667	7.89	1.61
15	16.667	66.667	16.667	7.69	1.7
16	16.667	16.667	66.667	7.83	1.66
17	100	0	0	8.41	1.74
18	0	100	0	7.97	1.82
19	0	0	100	8.27	1.82

**Table 6 materials-15-02470-t006:** Dissimilar weldment bead profiles data of TIG (conventional) and ATIG (optimal flux).

TIG			ATIG			
D	W	D/W	D	W_f_	W_b_	D/W
3.74	10.89	0.34	8.93	8.87	8.31	1.94

**Table 7 materials-15-02470-t007:** Ferrite volume proportions % in weld zone and HAZ for ATIG and TIG weldments.

Weld		Ferrite Volume Proportion %						
		Ferrite Volume in Different Locations in WZ				Average in WZ	HAZ 316L Side	HAZ 2205 Side
		1	2	3	4			
ATIG	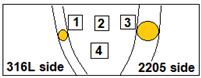	52	53	54	55	54	6.9	63
TIG	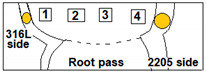	41	44	48	53	47	8	69

**Table 8 materials-15-02470-t008:** Measurements of tensile strength and standard deviation of TIG welds and ATIG welds with optimal flux.

Sample	Number of Tests	UTS Max. (MPa)	UTS Min. (MPa)	UTS Average (MPa)	Standard Deviations σ
316L SS BM	3	615	594	600	4
2205 DSS BM	3	820	796	815	11.5
TIG 2205 DSS/2205 DSS	3	800	791	795	4.73
TIG 316L SS/316L SS	3	598	594	596	2.08
TIG 2205 DSS/316L SS	3	597	591	594	1.00
ATIG 2205 DSS/316L SS	3	601	597	599	2.08

**Table 9 materials-15-02470-t009:** Measurements of hardness and standard deviation of TIG and ATIG (optimal flux) at FZ.

Sample	Zone of Tests	HV Max.	HV Min.	HV Average	Standard Deviations σ
TIG	FZ	242	232	237	2.96
ATIG	FZ	240	229	235	3.62

**Table 10 materials-15-02470-t010:** Measurements of hardness and standard deviation of TIG and ATIG with the optimal flux at HAZ.

Sample	Zone of Tests	HV Max.	HV Min.	HV Average	Standard Deviations σ
TIG	HAZ FZ/Duplex 2205SS	258	257	257.5	0.71
	HAZ FZ/316L SS	225	218	222	3.61
ATIG	HAZ FZ/Duplex 2205SS	258	255	257	2.12
	HAZ FZ/316L SS	210	209	210	1.53

**Table 11 materials-15-02470-t011:** Electrochemical data of ASS, DSS base metals, TIG, and ATIG weld metals obtained from potentiodynamic polarization studies in 3.5 M NaCl solution.

Sample No.	βc mV·dec^−1^	E_Corr_ mV	βa mV·dec^−1^	j_Corr_ µA·cm^−2^	E_pit_ mV SCE
DSS	314	−974	204	29.8	−434
ASS	288	−927	204	24.4	−455
WZ (TIG)	332	−1050	134	54.5	−383
WZ (ATIG)	331	−1032	170	54.1	−394
HAZ DSS (TIG)	337	−1012	149	31.3	−432
HAZ ASS (TIG)	236	−1021	124	37.9	−462
HAZ DSS (ATIG)	246	−1035	124	40.3	−323
HAZ ASS (ATIG)	422	−1005	187	38.7	−479

## Data Availability

The data used to support the findings of this study are included within the article.

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
