# Peer review of "Mechanical, Microstructure, and Corrosion Characterization of Dissimilar Austenitic 316L and Duplex 2205 Stainless-Steel ATIG Welded Joints"

_materials, 2022, doi:10.3390/ma15072470_

Round 1

Reviewer 1 Report

Review report on the topic ‘Mechanical, Microstructure and Corrosion Characterization of Dissimilar Austenitic 316L and Duplex 2205 Stainless Steels ATIG Welded Joints’. Comments are listed below:

  1. A common type of information was presented in the abstract section. The abstract represents the summary of the work. Revise it carefully.
  2. The novelty of the work should be discussed first in respect of the application.
  3. Try to strengthen the introduction section as many works have already been published on a similar topic. Try to make a bridge between current and previously published work and specify the gap area and objective of the work. Discuss the major problems associated with dissimilar joining, selection of filler metal, and the selection of PWHT temperature. Refer to the following: https://doi.org/10.3390/met10050559;
    https://doi.org/10.1007/s43452-021-00365-6;
  4. The literature section is very poor. Refer to some recently published work on dissimilar joining. Also, this is not a review paper, so please discuss in a paragraph and try to make a proper connection to reach a concrete research gap.
  5. How was the plate plate composition analyzed?
  6. Improve the quality of Fig. 2.
  7. Discuss in detail about A TIG process in the experimental section.
  8. Instead of Fig. 3 and Fig. 4 provide the image of the tensile and hardness specimen. Also, provide the image of the impact specimen along with notch location.
  9. In dissimilar welding, the major problem is about the interface. However, no line map was found across the interface. Improve the quality and if possible, add the line map to ensure the diffusion of the possible elements and also discuss the macrosegregation across the interface.
  10. Provide the image of the fractured specimen and add detail about the failure mechanism. If possible, add the results of BMs and compare the properties of the welded joint with BMs. Also mention about the joint efficiency and fracture location.
  11. Provide a detailed discussion about the fracture surface, which includes the dimples formation, cleavage area, and tear ridges. Also add the fractured specimen of impact tested specimen: doi:10.1016/j.engfailanal.2008.09.003.
  12. Shorten the length of the conclusion section.

Overall work is quite good and interesting and can be accepted after the following minor corrections.

Reviewer 2 Report

regarding the manuscript review entitled “Mechanical, Microstructure and Corrosion Characterization of Dissimilar Austenitic 316L and Duplex 2205 Stainless Steels ATIG Welded Joints,” the following points are provided.

  1. The manuscript lacks novelty and new scientific achievements.
  2. Analysis and communication between the contents are poor.
  3. The manuscript could not relate the effects of the applied active flux to the test results.

Therefore, the manuscript does not have the conditions for acceptance in the journal of materials.

For further explanation, I want to draw your attention to the following points (most issues were highlighted in the manuscript file).

  1. The manuscript has many scientific forms. For instance, at the end of the third paragraph, in lines 70-72, the definition of hot cracking is wrong.
  2. Tables 1 and 2 can be merged.
  3. The dimensions and thickness of both base metals must be clarified.
  4. In table2, the arc efficiency and Voltage of the welding process are not specified.
  5. The distance between micro-hardness indentations is long (inappropriate).
  6. 3.1, 3.1.1, 3.1.2, and 3.1.3 sections are unnecessary, and they can be added to materials and methods briefly.
  7. Root penetration of ATIG deposited weld metal is higher than 8 mm based on table5. Therefore, the weld is defective, and it is a significant problem.
  8. In section 3.2, it is claimed that the DSS2205 solidification mode is Ferritic-Austenitic. It is wrong because its ferritic solidification mode was well known. In addition, the determination of phases and solidification mode using WRC1992 is not well described.
  9. Description of charts an (ATIG) and b (TIG) do not match with figures subtitles.
  10. Figure 10 is not enough alone, and optical microscopy or EBSD should be performed.
  11. Determination of solidification mode needs WRC1992 and 50% dilution, and the method used in the manuscript is wrong.
  12. How has measured the ferrite content of the weld metal? It can be measured with WRC1992 and then FN measurement or using a ferrite scope instrument that is not clear in the text.
  13. Microstructural results have fundamental objections. It is unclear how the extension of the heat-affected zone (HAZ) has been measured. In addition, what is the criterion of distinction of HAZ and unaffected base metal?
  14. The first sentence of page 11 (line 338) is incorrect.
  15. The shown results in figures 13 and 14 are not citable because the linear or map analyses of weld metal toward base metal should be performed instead of point EDS analyses.
  16. The tensile test used to determine the tensile properties of weld joints is inappropriate. The notch tensile or nick-break test should be better for this case.
  17. a) The figure number must be changed to 8 from 7.
  18. b) The results do not conform with microstructural results.
  19. c) It is recommended that the results be illustrated as bar charts or curves.
  20. d) Figure 15 can be removed because it does not help the manuscript.
  21. e) The discussion and results in analyses are very poor.
  22. The effect of flux on microstructure and hardness must be added and discussed.
  23. Based on figure 16 and explanations in the text, it seems that tables 8 and 9 are not necessary.
  24. The discussion about impact test results is poor.
  25. It is recommended that table 10 be illustrated as bar charts.
  26. In the field of corrosion, it is recommended to perform cyclic polarization tests to obtain more information about the pitting corrosion of these alloys and their weldments.

Reviewer 3 Report

The article is devoted to an actual topic, namely the production of materials with high functional properties. In the article, the influence of the welding material (flux) on the properties of the welded joint of two types of corrosion-resistant steel grades 2205 and 316L is studied. The study uses modern research equipment, software and methods. The article is well-written, understandable and can be useful to readers. However, while reviewing the article, I noticed a few points that could be improved.

  1. In Figure 1, I recommend adding explanatory inscriptions. Specify the plates to be welded, flux, installation elements.
  2. Now Figures 1-5 are presented in low quality. I recommend to improve the quality of these photos and drawings.
  3. Section 2. «Materials» does not describe welding consumables (fluxes). I think that their characteristics should be added to this section.
  4. Information from 3.1.1 and 3.1.2, in my opinion, should be moved to Section 2. "Materials".
  5. Why is the use of an optical microscope CAROLINA not written in 2.4. Microstructure assessment?

Reviewer 4 Report

The paper ”Mechanical, Microstructure and Corrosion Characterization of Dissimilar Austenitic 316L and Duplex 2205 Stainless Steels ATIG Welded Joints” can be accepted in its present form.

This paper presents all the requested analyses in order to have a complete characterization.

-The results are interpreted appropriately
and all conclusions are justified and supported by the results.

-The article is written in an appropriate way.
Besides, the data and analyses are presented appropriately.

-The data are robust enough to draw the conclusions.
The methods, tools, software, and reagents are described with sufficient details to allow another researcher to reproduce the results.

In my opinion, the authors did a good job and their paper is suitable for publication. Maybe some figures need to be improved.

Round 2

Reviewer 1 Report

Accept.

Reviewer 2 Report

Unfortunately, this article lacks sufficient novelty and scientific achievement. So it does not seem acceptable.